# Characteristics and Service Utilization by Complex Chronic and Advanced Chronic Patients in Catalonia: A Retrospective Seven-Year Cohort-Based Study of an Implemented Chronic Care Program

**DOI:** 10.3390/ijerph18189473

**Published:** 2021-09-08

**Authors:** Sebastià J. Santaeugènia, Joan C. Contel, Emili Vela, Montserrat Cleries, Paloma Amil, Eva M. Melendo-Azuela, Esther Gil-Sánchez, Victoria Mir, Jordi Amblàs-Novellas

**Affiliations:** 1Central Catalonia Chronicity Research Group (C3RG), Centre for Health and Social Care Research (CESS), University of Vic—Central University of Catalonia (UVIC-UCC), 08500 Barcelona, Spain; sebastia.santaeugenia@gencat.cat (S.J.S.); jccontel@gencat.cat (J.C.C.); emelendo_ext@gencat.cat (E.M.M.-A.); 2Chronic Care Program, Department of Health, Generalitat de Catalunya, 08020 Barcelona, Spain; pamil@gencat.cat (P.A.); egils@gencat.cat (E.G.-S.); victoriamir@gencat.cat (V.M.); 3Unitat d’Informació i Coneixement, Servei Català de la Salut, 08006 Barcelona, Spain; evela@catsalut.cat (E.V.); mcleries@catsalut.cat (M.C.); 4Digitalization for the Sustainability of the Healthcare System (DS3), Sistema de Salut de Catalunya, 08006 Barcelona, Spain; 5Chair and Department of Palliative Care, University of Vic—Central University of Catalonia (UVIC-UCC), 08500 Barcelona, Spain

**Keywords:** chronic care, integrated care, geriatric care, palliative care, primary health care, multimorbidity, complexity, healthcare services utilization, complex needs, advanced chronic patients

## Abstract

The Chronic Care Program introduced in Catalonia in 2011 focuses on improving the identification and management of complex chronic (CCPs) and advanced chronic patients (ACPs) by implementing an individualized care model. Its first stage is their identification based on chronicity, difficult clinical management (i.e., complexity), and, in ACPs, limited life prognosis. Subsequent stages are individual evaluation and implementation of a shared personalized care plan. This retrospective study, including all CCPs and ACPs identified in Catalonia between 2013 and 2019, was aimed at describing the characteristics and healthcare service utilization among these patients. Data were obtained from an administrative database and included sociodemographic, clinical, and service utilization variables and morbidity-associated risk according to the Adjusted Morbidity Groups (GMA) stratification. During the study period, CCPs’ and ACPs’ prevalence increased and was higher in lower-income populations; most cases were women. CCPs and ACPs had all comorbidities at higher frequencies, higher utilization of healthcare services, and were more frequently at high risk (63% and 71%, respectively) than age-, sex-, and income level-adjusted non-CCP (23%) and non-ACP populations (30%). These results show effective identification of the program’s target population and demonstrate that CCPs and ACPs have a higher burden of multimorbidity and healthcare needs.

## 1. Introduction

Catalonia (Northeast of Spain) is one of the regions with the oldest population in the world due to its ever-increasing life expectancy and its lower fertility rate (83.5 years and 35.6%, respectively, in 2019) [1,2]. In 2016, public health expenditure per capita was 2137 (USD PPP), total health expenditure was 7.6% of the gross domestic product (GDP), and hospital beds supply was 1.7 per 1000, below the 3.7 per 1000 average of EU15. Currently, 19% of the Catalan population is aged >65 years, and this figure is projected to increase to >33%, with 12–15% of the population >80 years old by 2050. It is also one of the most intensively aging populations in the world [3,4]. This demographic evolution is associated with a higher prevalence of people with chronic diseases, who are currently a major healthcare and social concern, and constitute a burden for healthcare systems [5,6,7].

The national Catalan healthcare system provides free universal coverage, except for pharmaceuticals, which require a user co-payment, to a population of 7.6 million. One of its main distinguishing features is the separation of planning and financing functions, allowing for commissioning of healthcare services from public and private-owned centers, including acute care hospitals (67 centers and 12,776 beds), intermediate care hospitals (98 post-acute and long-term care centers with 8261 beds), mental health (40 acute mental health centers (3805 beds) and 129 community mental health network facilities) and primary care system facilities (377 primary care centers). The health plan is the main strategic planning instrument for all health interventions of the Government of Catalonia and, for the 2011–2015 period, was focused on improving care for patients who used it most: persons with chronic conditions and complex needs (CCPs) and those with advanced chronic conditions (ACPs) [8].

Compared with the general population, CCPs and ACPs have higher utilization of healthcare resources, including emergency admissions, consultations, and accident and emergency (A&E) services, higher drug use, and tend to be dependent [9,10]. However, healthcare systems are organized to treat singular diseases and, despite the increasing number of people with multiple chronic conditions and complex needs, still provide fragmented care [11,12]. As a result, standardized plans developed from a disease management perspective fail to fulfill CCPs’ and ACPs’ needs, increasing the risk of poor outcomes, such as emergency admissions, readmissions, and a higher number of primary healthcare visits [13]. Considering the concern raised by these unadjusted healthcare plans, new integrated and patient-centered care models have been developed worldwide, and these new interventions are being evaluated in other countries [12,14,15,16,17,18].

### 1.1. Patient Definition, Identification, and Specific Model of Care of CCPs and ACPs in Catalonia

To improve the main outcomes and care for CCPs and ACPs, the Catalan Chronic Care Program of late 2011 focused on developing a specific toolkit to define and identify CCPs and ACPs [10,16,19,20]. For the definition of CCPs, the Catalan program broadened the multimorbidity paradigm used in other regions of Spain and Europe and adopted people with complex care needs as population targets. Complexity was defined, following this new paradigm, in three dimensions, including clinical, social/contextual, and healthcare system complexities, and was treated as the result of the interaction between variables associated with each of the dimensions [21] (Figure 1).

Patients with chronic conditions and complex care needs (CCPs) were defined as those whose situation reflected the difficulty of their management and care and the need to adopt specific individual plans, owing to concurrent diseases, their utilization of healthcare services, and their context [22]. Complex care needs are the common feature among CCPs, who are estimated to account for 3.5–4% of the population [23]. The identification of CCPs is based on detecting relationships between different clinical, context-related, and health and social care system-related criteria. Clinical criteria include multimorbidity, dynamic and unpredictable outcome, and classification within the 5% at higher risk according to the Adjusted Morbidity Groups (GMA, from Spanish “Grupos de Morbilidad Ajustados”) model. Context-related criteria include dysfunctional or risky social situations to meet the person’s needs, and health and social care system-related criteria include, among other factors, patient management differences among each healthcare professional settings. Unlike specific screening instruments, a specific, unequivocal guideline to define CCPs has not been established. Despite the lack of a perfect algorithm to identify CCPs, a perceptive decision related to complexity criteria has been established as the basis of the model and is used by healthcare professionals to define a patient as CCP. The definitions of complexity and their associated criteria are summarized in Appendix A.

In addition to complex care needs, people with advanced chronic conditions (ACPs) need palliative care and have a limited life prognosis (from a few days up to one year). Considering the previous evidence regarding the benefits of early identification and palliative care in this population (i.e., first end-of-life transition) [24], the Catalan healthcare plan has changed its paradigm of care for people needing palliative care from the classical dichotomic perspective (i.e., curative vs. palliative care) applicable to cancer patients in their last days or weeks in palliative care units, to a dynamic synchronous perspective applicable to any disease and advanced chronic condition, regardless of the person’s location [25]. This view of palliative care is different compared to other countries and seeks to identify patients in end-of-life transition needing early palliative care. To facilitate the early identification of ACPs in Catalonia, the Department of Health and the Chair of Palliative Care of Vic University developed a screening tool for the early identification of the need for palliative care among individuals with limited life expectancy: the NECesidades PALiativas tool (NECPAL-CCOMS-ICO) [26]. The NECPAL tool is a validated instrument to screen and identify people with palliative care needs that combines the surprise question (‘Would you be surprised if this patient dies in the next year?’) with other items, including the request for palliative care by the patient or family and the need for palliative care as identified by professionals, general clinical indicators, psychosocial factors, multimorbidity, use of resources, and specific indicators to evaluate disease severity and progression. Previous studies in the Catalan setting have shown a 1–1.5% prevalence of APCs, with advanced frailty and/or dementia (55%), advanced organ disease (32%), and cancer (13%) [26].

Providing optimal care to CCPs and ACPs requires combining the points of view of healthcare systems, responsible for facilitating a better response to these patients through healthcare planning and resource management, and healthcare professionals responsible for providing care based on the multidimensional needs of people. Furthermore, success relies on incorporating the population view, based on care and organizational models to respond to the needs of this population, and the individual view, based on patient-centered models providing individualized care, considering that the main goal is to obtain good results from the patients’ perspective [27]. Along these lines, the Department of Health of Catalonia developed the Chronic and Integrated Healthcare program, an individualized care model that integrated the population perspective, aimed at improving provision of services and integrating care, which revolves around primary care teams, with the individual perspective, aimed at customizing care for each individual patient (Figure 2). This individual care model included four stages, of which patient identification as CCPs and ACPs is the first.

### 1.2. Justification and Aims of This Study

Even though the Chronic and Integrated Healthcare program has been implemented for several years, the populations identified as CCPs and ACPs remain to be analyzed. Furthermore, few studies have evaluated other initiatives aimed at providing integrated care for people with chronic conditions and complex needs, often with unexpected results [28]. The healthcare system of Catalonia registers patient stratification based on the GMA categories and identification as CCP or ACP on electronic healthcare records, making this information available for its monitorization and evaluation. In this study, we used the Catalan electronic administrative clinical database to describe the sociodemographic and clinical characteristics of patients identified as CCPs and ACPs with the goal of assessing the first stage of the Chronic Care program in the first few years of its implementation. Additionally, we evaluated the clinical characteristics and utilization of healthcare services among these populations.

## 2. Materials and Methods

### 2.1. Study Design, Participants, and Database

This was a retrospective analysis of an administrative database that included all individuals identified as CCPs or ACPs in Catalonia (northwest of Spain) between 2013 and 2019. CCPs and ACPs were identified by primary care specialists in 377 primary care centers based on the criteria previously presented (Appendix A). Owing to the use of an electronic database as the data source and the irreversible anonymization of the data extracted, patient informed consent was not applicable in this study. This study was conducted in accordance with the Ethical Principles for Medical Research Involving Human Subjects of the Helsinki Declaration and the local Personal Data Protection Law (LOPD 15/1999); it was approved by the Research Ethics Committee of University of Vic/Central University of Catalonia (UVIC-UCC) reference number 63/2018.

### 2.2. Data Source

Sociodemographic and clinical data were obtained from the Catalan Health Surveillance System (CHSS) that, since 2011, has been collecting detailed information about the utilization of healthcare by/among the entire population of Catalonia (7,600,000 inhabitants). This record, which has been analyzed in previous publications in other areas [29,30,31], gathers data recorded in multiple settings, including primary care, acute care hospitals, intermediate care hospitals, mental health centers, outpatient clinics, and emergency services. Furthermore, this record collects information regarding prescriptions and pharmacy expenses and invoices, including outpatient clinics, non-urgent medical transportation, outpatient rehabilitation, home oxygen therapy, and dialysis. No data about private healthcare could be collected because these centers use different codes for patient identification.

### 2.3. Variables

The sociodemographic variables considered in this study were age, sex, and income level, classified as high (annual income >100,000 €), intermediate (18,000–100,000 €), low (<18,000 €), and very low (receiving welfare support from the government). Clinical variables were diagnoses, as they appear in the CHSS database according to the usual clinical practice, and coded according to the International Classification of Diseases, ninth revision, Clinical Modification (ICD-9-CM). The multimorbidity burden was stratified based on the Adjusted Morbidity Groups (GMA), which considers the type of disease—acute or chronic—, number of systems affected, and complexity of each disease [32,33]. The GMA enable the classification of all the population into four strata based on their morbidity-associated risk. The four strata are (1) Baseline risk (healthy stage), with an GMA score up to the 50th percentile of the total population; (2) Low risk, with a GMA score between the 50th–80th percentiles; (3) Moderate risk, with a GMA score between the 80–95th percentiles; and (4) High risk, with a GMA score above the 95th percentile [32,34]. Variables associated with the utilization of healthcare services during the first year after identification of CCPs and ACPs were number of (1) visits to primary healthcare centers; (2) outpatient visits; (3) emergency service admissions; (4) acute care hospital admissions and length of stay (days); (5) admissions to intermediate care hospitals and length of stay (days); (6) admissions to psychiatric centers and length of stay (days), and (7) prescribed drugs, according to the different chemical and therapeutic classification groups, and units.

In order to describe the evolution of epidemiological and clinical characteristics among CCPs and ACPs, an annual incidence study was conducted. Health expenditure was calculated according to the standard costs of each service provided by the Department of Health (Generalitat de Catalunya) for each year [35].

### 2.4. Statistical Methods

Categorical variables were described as frequencies and percentages and quantitative variables as the mean and standard deviation (SD) and/or the median and interquartile range (IQR; Q1, Q3). Incidence and prevalence rates were expressed per 1000 inhabitants and mortality rates per 100. Categorical variables were compared using Pearson’s Chi-squared test with Yates’ continuity correction. Survival curves were calculated using the Kaplan–Meier estimator and compared using the Gehan test. The statistical significance threshold was set at a bilateral alpha value of 0.05. The utilization of healthcare services and associated expenditure of CCPs and ACPs were compared with the population of patients not identified as complex adjusted by age, sex, and income level, hereinafter referred to as “non-CCP” and “non-ACP”, respectively. Comorbidities and healthcare services utilization were compared using the rate ratio by median-unbiased estimation (mid-p), and healthcare services expenditure was compared using the Student’s *t*-test. To analyze geographic variability, the Poisson regression was used to calculate cumulative incidence rates for the 2017–2019 period, adjusted by age, sex, morbidity (GMA), and income level. Data after the first few years of implementation of the Chronic Care program (2017–2019) were considered to be more stable and were aggregated to increase the robustness of these analyses. All analyses were performed using the R statistical package (version 4.0.3).

## 3. Results

### 3.1. Characteristics of the Overall Cohort (2013–2019)

During the study period, 303 357 individuals with a median (IQR) age of 82 (74.0, 86.0) years were identified as CCPs, and 98,587 persons with a median (IQR) age of 84.0 (75.0–90.0) years were identified as ACPs. Table 1 summarizes the sociodemographic and clinical characteristics of the overall study population according to their identification as CCP and ACP. Sociodemographic and clinical characteristics of CCPs and ACPs identified throughout the study period (2013–2019) were significantly different between groups, although differences were small. ACPs were older and, as expected, were more frequently classified in the high-risk GMA stratification category compared to CCPs. Dementia was more frequent in ACPs compared to CCPs.

Survival analysis showed a significantly decreased probability of survival of ACPs compared to CCPs (Figure 3). One-, 3-, and 5-year survival rates were 82.4% and 37.0%, 58.2% and 14.8%, and 39.3% and 5.4% for CCP and ACP, respectively.

### 3.2. Epidemiological Evolution of the Identification of CCPs and ACPs

During the study period (2013–2019), the prevalence rates of both CCPs and ACPs in the general population increased from 8.8 and 1.2 cases per 1000 people in 2013 to 21.7 and 2.6 cases per 1000 people in 2019, respectively. Conversely, incidence rates decreased from 2013 to 2019 in both populations, from 9.1 to 3.9 new cases per 1000 individuals for CCPs and from 1.8 to 1.6 new cases per 1000 individuals for ACPs, respectively. CCP prevalence and incidence rates were more variable during the study period compared with those of ACP (Figure 4).

Analysis of survival rates according to the year of identification showed significantly and progressively decreased survival of ACPs and, even though survival curves of CCPs showed a similar significant trend, differences between years were more modest (Figure 5).

From the beginning and until the end of the study period, the mean age of CCP and ACP incident cases increased in men and women. Mean ages of patients identified as CCP and ACP were persistently higher in women throughout the whole study period: Age differences ranged from 3.8 to 4.3 years for CCP and from 4.9 to 5.7 years for ACP. Sociodemographic and clinical characteristics of patients identified as CCP and ACP showed significant changes throughout the study period. The ages of patients identified as CCPs and ACPs increased gradually and slowly (1.5-year and 1.7-year differences for CCPs and ACPs, respectively, in 7 years), and the proportion of patients at a high and very high GMA risk progressively increased, with a concomitant decrease in patients at low and moderate risk. The prevalence of different morbidities throughout the study period significantly changed (Table 2 and Table 3).

### 3.3. Evaluation of Demographic and Clinical Characteristics of CCPs and ACPs (2019)

In 2019, the total number of CCP cases (prevalence) was 167,892, of which 98,676 were women and 69,216 were men. The prevalence of ACP was lower, with 19,741 individuals, of which 11,907 were women and 7834 were men. The distribution of these populations by age and gender is shown in Appendix A.

Regarding the distribution of CCPs and ACPs according to socioeconomic level and sex, the prevalence of both CCPs and ACPs progressively increased as the socioeconomic level decreased for both women and men, with an overall higher prevalence of both CCPs and ACPs in the low- and very low-income categories. The population of women with very low and low-income had the highest prevalence of CCPs and ACPs, respectively (Table 4).

The clinical characteristics of CCPs and ACPs were compared to those of adjusted non-CCP and non-ACP populations. According to the GMA stratification of the morbidity-associated risk, most CCP and ACP cases were at high risk, representing an increased proportion of patients in this risk level compared with their respective adjusted non-CCP and non-ACP populations (63% vs. 23% for CCP and 71% vs. 30% for ACP, respectively) (Figure 6).

Accordingly, all comorbidities were present at significantly higher frequencies in CCPs and ACPs compared with their adjusted non-CCP and non-ACP populations (Table 5). The most frequent comorbidity in CCP patients was diabetes, followed, in this order, by chronic kidney disease, heart failure, cancer, and chronic obstructive pulmonary disease (COPD), whereas in ACPs, cancer was the most frequent comorbidity, followed by chronic kidney disease, heart failure, dementia, and diabetes. The distribution of morbidities by sex is shown in Appendix A.

### 3.4. Evaluation of Health Service Utilization and Associated Expenditures of CCPs and ACPs (2019)

Table 6 summarizes the utilization of healthcare services by CCPs and ACPs and their associated expenditure. Compared with their adjusted non-CCP and non-ACP populations, CCPs and ACPs had significantly higher utilization of the different healthcare services, including primary care, outpatient care, emergency admissions, day hospital, and mental health, and were prescribed a higher number of drugs.

Admission rates in acute care hospitals, intermediate care hospitals, and psychiatric centers were also higher in CCPs and ACPs than in their respective non-CCP and non-ACP populations. Differences with their corresponding age, sex, and income level-adjusted non-CCP and non-ACP populations were particularly higher for utilization of mental health services and admission to psychiatric centers in CCPs and day hospital and intermediate care hospital admissions in ACPs. Expenditures derived from primary care, outpatient care, hospital admissions, emergency departments, mental health, intermediate care hospitals, prescribed drugs, and other healthcare services were also significantly higher for CCPs and ACPs compared to the non-CCP and non-ACP populations adjusted by age, sex, and income level, with the exception of ACPs’ mental health services expenditure. Accordingly, total expenditures on healthcare services were substantially and significantly higher for CCPs and ACPs than for their corresponding adjusted non-CCP and non-ACP populations (Table 6). While hospital admissions and prescribed drugs were the main expenses in all patient groups (i.e., CCPs, ACPs, and the corresponding non-CCP and non-ACP populations), differences in expenditures associated with mental health and outpatient care between CCPs and ACPs and their corresponding non-CCP and non-ACP populations, respectively, were higher than those of other expenditures.

### 3.5. Geographical Variability of CCPs and ACPs Incidence in Catalonia

CCPs’ and ACPs’ cumulative incidence for the 2017–2019 period in the different regions of Catalonia was adjusted to the population’s age, sex, morbidity, and income level, revealing regions with increased and decreased case incidence compared to the expected rates (Figure 7). Overall, adjusted incidence indexes of CCPs and ACPs were similar in each individual region.

## 4. Discussion

Owing to the healthcare needs and service utilization rates of individuals with chronic conditions and complex needs, the health plan of the Government of Catalonia for the 2011–2015 period prioritized this highly demanding population for the implementation of the chronic care program. To provide them with the best possible care, the Catalan Department of Health developed an integrated and individualized model of care structured in four stages, of which the first entails the screening and identification of CCPs and ACPs. In the healthcare system of Catalonia, the GMA stratification category and identification as CCP and ACP are registered on patients’ medical records, enabling the use of this information. This observational, retrospective study, including all CCPs and ACPs from Catalonia identified between 2013 and 2019, assessed the first stage of this model during its initial implementation. During the study period, prevalence and incidence rates of CCPs and ACPs increased and decreased, respectively, while the probability of survival was significantly lower in ACPs compared to CCPs, and progressively decreased in both groups. The evolution of prevalence and incidence rates and survival probabilities are compatible with the progressive identification of CCPs and ACPs during the initial implementation of a novel, innovative care model. Likewise, the overall sociodemographic and clinical characteristics of CCPs and ACPs significantly changed. The prevalence of both CCPs and ACPs was higher in populations with low and very low income, and most cases were women. Compared to their respective age-, sex-, and income level-adjusted non-CCP and non-ACP populations, CCPs and ACPs were at higher morbidity-associated risk, had higher rates of all comorbidities, and higher utilization of healthcare services and associated health expenditure.

This observational study analyzed the implementation of a novel, innovative healthcare model centered on persons with complex conditions and care needs and advanced chronic diseases. Specifically, this study assessed the identification of the model’s target population during the first years of implementation. The initial stages of the implementation of a novel model are likely associated with a learning curve for the identification of both CCPs and ACPs, and increased efforts to implement the program among physicians, which may explain the progressively decreased case incidence and increased case prevalence. In this regard, prevalence rates are expected to increase as the model is consolidated and reaches expected rates [23]. The incidence and prevalence rates found throughout the study revealed a faster identification of CCPs compared to ACPs. In this regard, given that patient identification relies on physicians’ criteria and expertise applying the established perceptive criteria, effective identification is likely to be associated with a learning curve, even though GMAs have been published in the eHealth records since their introduction. Additionally, considering the time during visits required for identification, the substantial burden of primary care teams in our setting may have likely impacted the identification of CCPs and ACPs. Other shortcomings of this model are associated with ethical discussions during the first years, likely leading to a reluctance by some clinicians to identify these populations, particularly ACPs, given their palliative connotation. Ethical issues are particularly relevant in the case of ACPs, given the previously acknowledged prejudices and fears towards the identification and care of people needing palliative care, who may not receive appropriate care in case of a health crisis [36]. Furthermore, clinicians who need clear rules and guidelines may regard the need to combine objective and professional (subjective) perceptions of complexity as a barrier for identification, further contributing to increased reluctance. Future interventions from the Department of Health should focus on highlighting the benefits of early identification of the model’s target population to overcome these shortcomings and implement an individual intervention plan shared among clinicians (second and third stage of the integrated care model), as depicted in Figure 2. In this respect, the patient identification information is accessible and visible to all the healthcare system, including acute and intermediate care centers, mental health centers, emergency services, and, in certain territories, social services. Despite the model’s shortcomings, most CCP and ACP prevalent cases were identified during the first years. In this regard, in the first years of implementation of the model (2013–2017), identification was encouraged with economic incentives to clinicians, partly explaining the decreased identification after this period. The economic incentives were objectives introduced in the commissioning process and incorporated a variable pay based on the achievement of goals. Despite expediting identification, economic incentives raised clinical and ethical controversies among professionals, and, currently, quality (i.e., added value to the patient) of identification is prioritized over quantity. Case identification (i.e., incidence of CCPs and ACPs) peaked during the first years (in 2014 for both CCPs and ACPs) and decreased from 0.96% and 0.25% in 2014 to 0.39% and 0.16% in 2019, respectively. The higher incidence rates observed during the first years are compatible with the implementation of the first stage of the novel model and reflect the effective identification of CCPs and ACPs.

ACPs had a lower probability of survival, consistent with their identification as patients with low life expectancy prognosis, indirectly confirming the validity of ACP definition [16]. Regarding the prevalence of ACPs, previous cross-sectional studies using validated tools (i.e., NECPAL) to identify patients with chronic conditions in need of palliative care (similar to the ACP definition) reported rates of 1–1.5%, higher than the 0.26% found in this study [25,26]. However, whereas these previous studies aimed at prospectively identifying these patients, this study was conducted in a real-world setting and reflected the heterogeneity among clinicians, similar to the geographical variability, likely explaining the observed differences. As explained earlier, this heterogeneity may be related to ethical issues associated with the identification of ACPs. Furthermore, the reluctance of some clinicians to identify CCPs as ACPs may have additionally contributed to these discrepancies. In this regard, clinicians have shown increased reluctance to use the ACP identification, likely resulting in decreased identification of this subgroup of CCPs. Nevertheless, the decreased prevalence of ACPs compared to that estimated in previous cross-sectional studies (20% of estimated ACP prevalence) and the limited availability of international experiences in the field of chronicity focused on the proactive identification of ACPs warrant further research [25,26].

The use of the concepts of complexity and multimorbidity to define and identify CCPs and ACPs is novel and unique in this model. The concept of complexity in the context of healthcare lacks a precise definition and, in addition to clinical factors (i.e., chronic diseases), it encompasses other patient-related factors (i.e., socioeconomic), physician-related factors, including training, expertise, and experience, factors related to the organization of care, including decision-making, workflow, technology, and availability of time, team-related factors (i.e., leadership), contextual factors (physical and social), and organizational factors, including structures, politics, and procedures [21,37,38]. Unlike clinical variables, routine electronic clinical records do not systematically record most social factors and clinical fragmentation variables that determine the complexity and, overall, the availability of structured information regarding social variables is limited. Given the diversity of constructs that have been associated with complexity, an international consensus on its definition is needed to homogenize results from different studies and understand the care needs of complex patients [20,39]. In contrast, the availability of validated screening instruments enabled the identification of ACPs based on a robust construct. In our setting, the NECPAL tool, a validated instrument for the early identification of the need for palliative care among individuals with limited life expectancy, is routinely used [26,40,41]. Despite differences in the application of the concept of palliative care among countries (i.e., patients with oncologic conditions and in the last weeks or days of life vs. management of advanced chronic conditions), the definition of ACPs included in the Catalan healthcare plan used a robust method for identification, similar to other countries, potentially enabling comparisons among different countries/settings.

While the identification of CCPs depended on professionals’ subjectivity (i.e., perception) regarding the concept of complexity, which was supported by information communication technology tools for stratification, the initial screening considered unique functional identifications related to patients’ complexity status, regardless of the number of chronic conditions, using the automatic and hence, objective, GMA stratification system [16,32]. A complexity status detected by the initial GMA stratification system is likely to be associated with difficult management and decision-making. In this context, the GMA algorithm is a useful non-invasive support stratification tool for the initial identification by primary healthcare teams of people with potentially complex healthcare needs, candidates to be identified as CCPs, and to whom the integrated and individualized healthcare model developed by the Department of Health is applicable. This screening allows labeling patients and prioritizing them for their subsequent evaluation and identification as CCPs and ACPs. The results obtained regarding their demographic and clinical characteristics and their healthcare services utilization using the support stratification method (i.e., GMA) support the validity of the CCP and ACP constructs defined in the Integrated Chronic Care program to identify patients with specific care needs.

Analysis of the prevalence of comorbidities revealed that all of them were more frequent in CCPs and ACPs than in age-, sex-, and income level-adjusted non-CCP and non-ACP populations, showing an increased morbidity burden. Despite ranking in similar positions regarding frequency, cancer ranked fourth in CCPs (34.2%) and was the most frequent comorbidity in ACPs (48.5%) and, conversely, diabetes ranked as the most frequent comorbidity in CCPs (43.8%) and fifth in ACPs (38.3%), showing trends consistent with patients’ end-of-life situation. In this regard, while most patients needing palliative care identified in previous studies using the NECPAL tool were in the dementia trajectory (55%), in this study, cancer was the most frequent comorbidity in patients identified as ACP. Even though patients with complex statuses included in this study may only have one chronic disease, considering the fact that most people with chronic disease have multimorbidity and the overall high prevalence of chronic conditions in CCPs and ACPs, most of this study’s population likely had multimorbidity [42]. Regardless of the frequencies of comorbidities and the number of chronic diseases, CCPs and ACPs were at substantially higher morbidity-associated risk.

Despite potential differences in the definition of complexity, patients with multimorbidity and functional limitations have higher needs compared to multimorbid patients [43]. Accordingly, compared with age-, sex-, and income-level-adjusted non-CCP and non-ACP populations, CCPs and ACPs had substantially higher utilization of healthcare services. Primary care and acute care hospital admissions were the most frequently used. Furthermore, low- and very low-income population segments had a higher prevalence of CCPs and ACPs, indicating a relationship between socioeconomic and complex chronic statuses, similar to previous studies showing relationships between multimorbidity and income and educational levels [42,44,45,46]. In this regard, patients were classified according to income arbitrarily using data available from pharmacy records. These classification criteria used ad hoc precluded comparisons with other studies. The previously reported relationships between low income and more intensive use of primary care and high income and higher use of specialists and the higher prevalence of CCPs and ACPs in populations with lower income may explain their higher use of primary care services [47]. The substantial increase in the use of all healthcare resources and their associated expenditures in CCPs and ACPs compared to their corresponding adjusted non-CCP and non-ACP populations underscores the anticipated impact of complex chronic patients on the healthcare system.

The healthcare system of Catalonia uses a stratification algorithm (i.e., GMA) and specific identifiers for CCPs and ACPs, which are registered on the electronic health records, allowing us to monitor and use this information. To our knowledge, the availability of this information is unique to the Catalan healthcare system or is at least very rare in other settings. Furthermore, the tools (i.e., information system data and individual patient assessment) and criteria (i.e., clinical, context-related, and health and social care system-related) used for CCPs and ACPs identification are unique of the Chronic Care program precluding direct comparisons with previous reports. Previous studies have described similar populations using the high needs, high costs concept, corresponding to those patients who use the healthcare system the most. This criterion is typically used to define and identify patients with multiple chronic conditions [8]. A meta-analysis of studies evaluating patients with high needs, high costs showed that similar to this study, these patients had increased healthcare resource utilization, were more likely to die, and their most frequent comorbidities were similar to those of CCPs and ACPs [48]. Additionally, both social and material deprivation (similar to the context-related criteria considered in this model) were associated with higher costs [48]. However, the high needs, high costs patients identified in these previous studies using information system data were younger than the population identified in this study: half were younger than 65 years, whereas in this study, only 10.4% and 10.3% of CCPs and ACPs, respectively, were within this age range [48].

Regarding the prevalence of patients equivalent to CCPs, the previously reported prevalence rates of patients with multimorbidity differed across studies and settings [49]. Overall, prevalence rates in low- and middle-income countries were lower compared to those in high-income countries. Regarding ACPs, several previous models aiming to identify persons needing palliative care have been evaluated, but specific data regarding the prevalence of ACPs was not reported [50].

The results of this study should be interpreted in the context of some methodological limitations associated with its retrospective design and real-world setting, including variability in recorded data and the tools used for patient identification. In this regard, the GMA algorithm considers clinical variables of chronic diseases to measure the morbidity burden, and clinicians’ subjective criteria are fundamental to evaluate other areas of complexity excluded from the GMA, including classification of chronic diseases according to severity and stages. In this regard, additional tools measuring the social- and healthcare system-related complexity dimensions are needed to gather structured, good-quality data on social variables, such as dependency, poverty, poor housing, and loneliness, beyond the management perspective of current algorithms. As structured variables become progressively available, they will be incorporated in the assessment of the different complexity dimensions. An additional limitation of this study is related to the identification of ACPs and their lack of validation on a case-by-case basis. Given the ethical issues associated with ACP identification, some clinicians may be hesitant to identify patients as ACP, potentially resulting in inaccurate ACP identifications of some cases. In this regard, future studies aimed at assessing healthcare service utilization and characteristics of the ACP population may require a prospective design to ensure data reliability. Nevertheless, the use of a large dataset including all people using the public healthcare system lacked selection bias and likely compensated for missing data and potential inaccuracies, at least partly, allowing us to capture the characteristics of chronic complex patients at the population level. The classification used in this study is unique to the Catalan Healthcare system. Therefore, the results of this study may not be applicable to other countries using other screening tools, identifiers, and other strategies to manage patients with multimorbidity and those with a short life prognosis [11,51,52]. Furthermore, the results from this model are unlikely to be applicable to healthcare systems of developing countries with poorly established primary healthcare systems.

Despite these limitations, this study assessed basic demographic and health indicators, allowing the characterization of the population with chronic complex conditions and describe their evolution in the context of the Chronic and Integrated Care Program. Future studies should focus on assessing trends in healthcare service utilization and expenditures to assess the impact of the Integrated Chronic Care Program in the management of these patients and their outcomes. Nevertheless, the results from this and future studies will be very useful to identify the challenges of implementing an integrated care model by the Department of Health. Future studies should assess other patient-related factors, such as patient experience outcome measures (PREMs), satisfaction, self-perceived health, and early access to palliative care, which may influence health-related quality of life, survival, healthcare costs, and end-of-life care, to ultimately improve the Integrated Care Program.

## 5. Conclusions

In the framework of the Chronic and Integrated Healthcare Program, the target populations of CCPs and ACPs were effectively identified, revealing their higher prevalence in low- and very low-income populations. CCPs and ACPs showed a higher frequency of multimorbidity, morbidity-associated risk, and utilization of healthcare services compared with the population of the same age, sex, and income level, reflecting their higher needs and expenditure. These results underscore the need to provide integrated care to complex chronic patients from the healthcare and social perspectives to improve and optimize their management. In the context of the increasing prevalence of people with complex chronic conditions, strategies, such as the Chronic Care Plan assessed in this study, which focus on this population of patients, should be implemented and assessed with the goal of decreasing their burden on the healthcare system.

## Figures and Tables

**Figure 1 ijerph-18-09473-f001:**
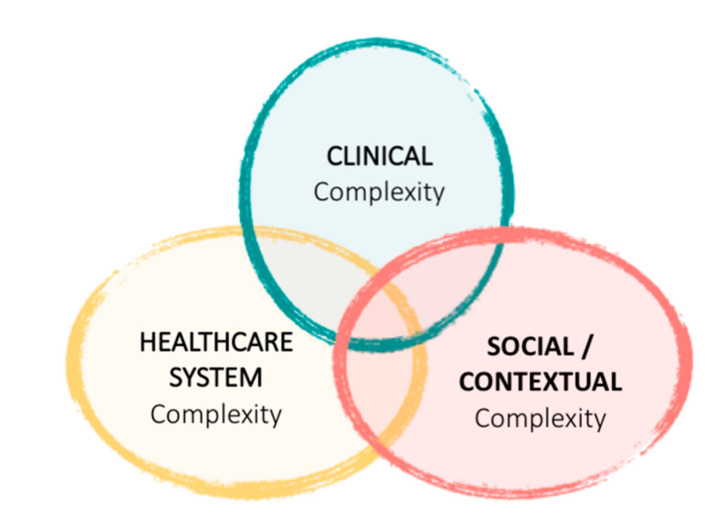
Dimensions of complexity. Adapted from de Kuipers et al. [21].

**Figure 2 ijerph-18-09473-f002:**
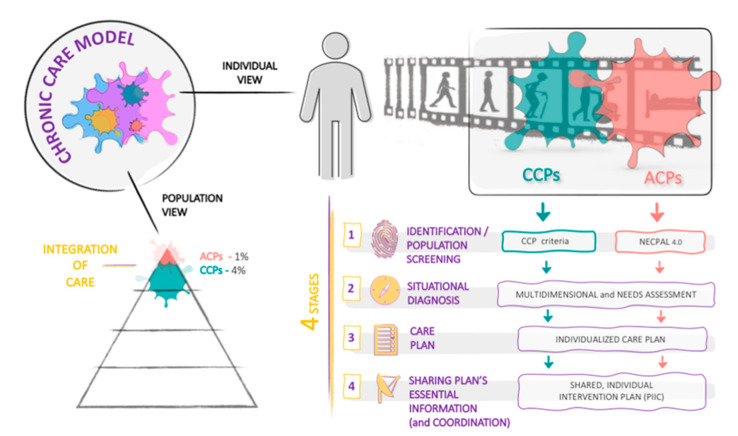
Diagram depicting the model of care for persons with complex chronic conditions and advanced chronic conditions of the Catalan health program 2011–2015 from the individualized and population perspectives. The model is organized in four main stages. ACPs, advanced chronic patients; CCPs, complex chronic patients.

**Figure 3 ijerph-18-09473-f003:**
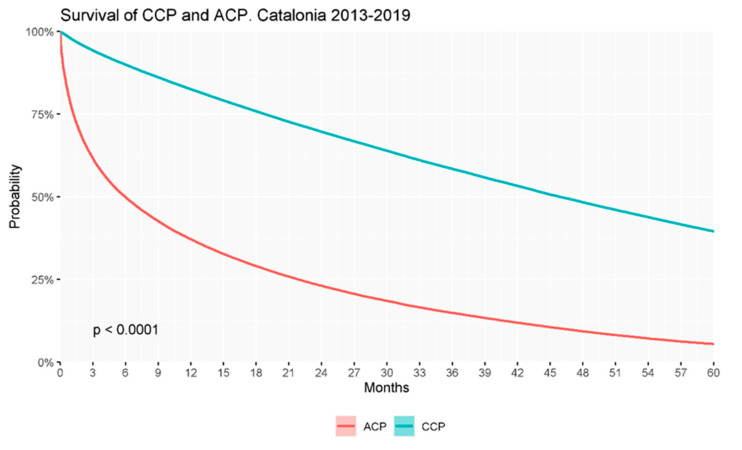
Probability of survival up to 5 years of the overall population of chronic complex patients and advanced chronic patients identified throughout the study period (2013–2019).

**Figure 4 ijerph-18-09473-f004:**
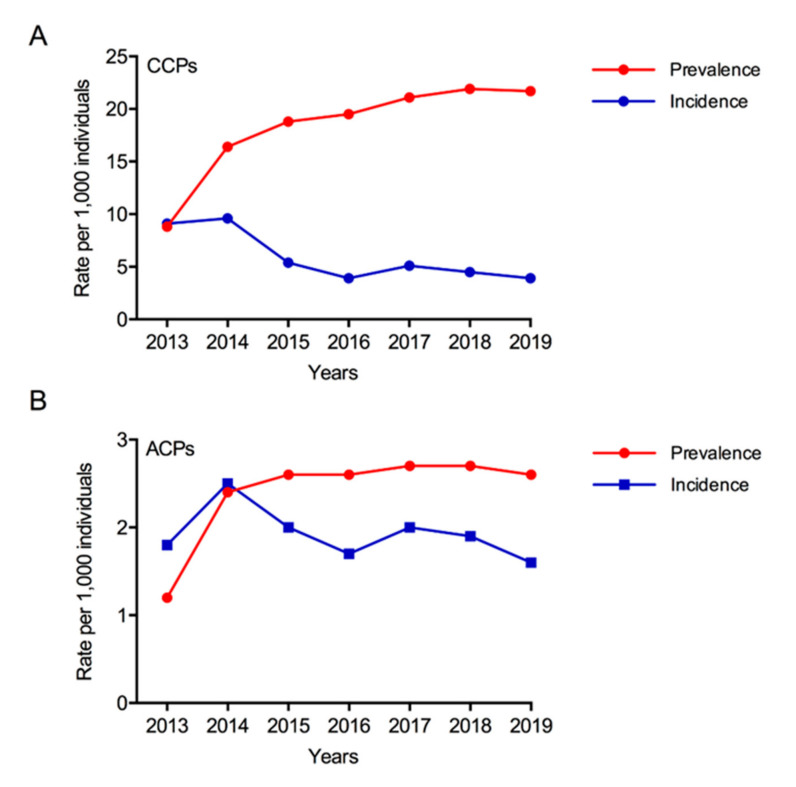
Prevalence and incidence rates of (**A**) CCPs and (**B**) ACPs throughout the study period (2013–2019).

**Figure 5 ijerph-18-09473-f005:**
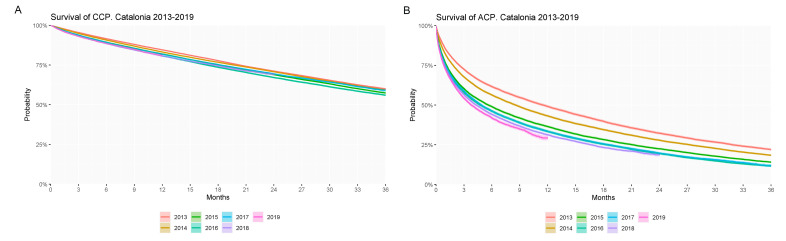
Probability of survival up to 3 years of (**A**) chronic complex patients (CCPs) and (**B**) advanced chronic patients (ACPs) according to year of identification.

**Figure 6 ijerph-18-09473-f006:**
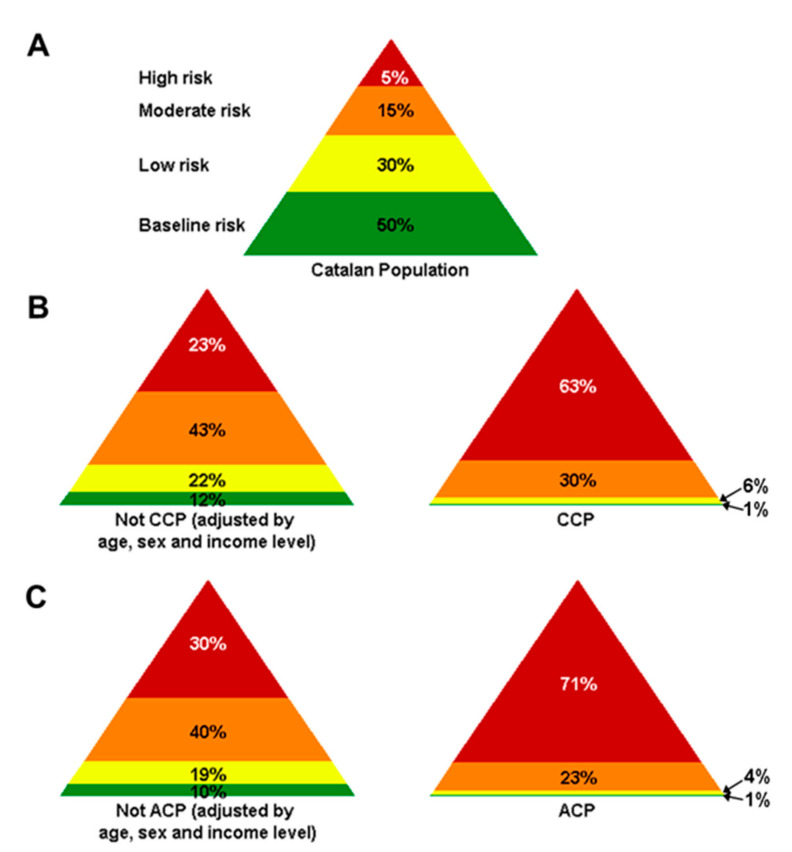
GMA stratification according to comorbidity risks of (**A**) total Catalan population, (**B**) CCPs, and (**C**) ACPs and their corresponding age-, sex-, and income level-adjusted non-CCP and non-ACP populations in 2019.

**Figure 7 ijerph-18-09473-f007:**
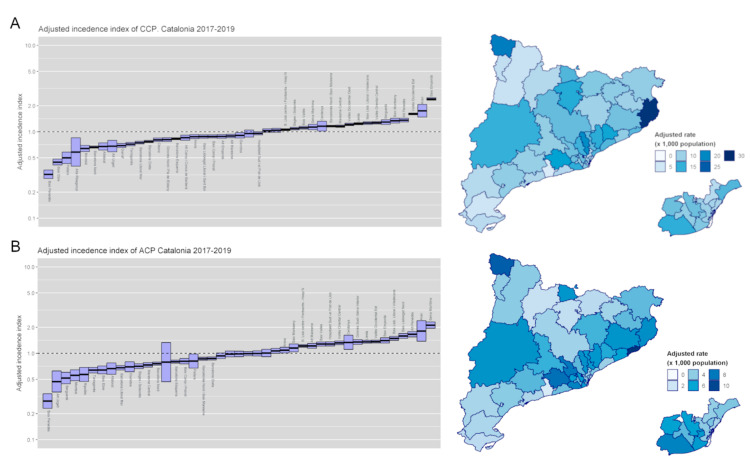
Incidence of (**A**) chronic complex patients (CCPs) and (**B**) advanced chronic patients (ACPs) adjusted by age, sex, comorbidities, and income level in the different regions of Catalonia in 2019.

**Table 1 ijerph-18-09473-t001:** Sociodemographic and clinical characteristics of the overall study cohort (2013–2019) according to their identification as complex chronic patients and advanced chronic patients, *n* (%).

	CCP	ACP	*p*-Value ^a^
	N = 303,357	N = 98,587	
**Sociodemographic characteristics**			
Sex			
Male	133,454 (44.0)	46,007 (46.7)	<0.001
Female	169,903 (56.0)	52,580 (53.3)	
Age, years			
<15	1020 (0.34)	134 (0.14)	<0.001
15–44	5060 (1.67)	1074 (1.09)	
45–64	25,466 (8.39)	8921 (9.05)	
65–74	45,554 (15.0)	12,871 (13.1)	
75–84	113,495 (37.4)	28,701 (29.1)	
>84	112,762 (37.2)	46,886 (47.6)	
Income level			
High	701 (0.23)	350 (0.36)	<0.001
Medium	46,586 (15.4)	16,439 (16.7)	
Low	244,127 (80.5)	78,429 (79.6)	
Very Low	11,930 (3.93)	3358 (3.41)	
**Clinical Characteristics**			
GMA stratification			
Baseline risk	1704 (0.56)	304 (0.31)	<0.001
Low risk	19,826 (6.54)	3508 (3.56)	
Moderate risk	119,234 (39.3)	26,512 (26.9)	
High risk	162,593 (53.6)	68,263 (69.2)	
Comorbidities			
Arterial hypertension	247,001 (81.4)	76,284 (77.4)	<0.001
Arthrosis	157,006 (51.8)	47,320 (48.0)	<0.001
Diabetes mellitus	124,762 (41.1)	35,848 (36.4)	<0.001
Heart failure	100,330 (33.1)	35,084 (35.6)	<0.001
Chronic kidney disease	98,985 (32.6)	34,539 (35.0)	<0.001
Chronic obstructive pulmonary disease	87,748 (28.9)	29,954 (30.4)	<0.001
Depression	91,961 (30.3)	28,305 (28.7)	<0.001
Ictus	71,847 (23.7)	25,735 (26.1)	<0.001
Ischemic heart disease	70,247 (23.2)	21,187 (21.5)	<0.001
Dementia	43,957 (14.5)	20,492 (20.8)	<0.001
Osteoporosis	53,707 (17.7)	16,572 (16.8)	<0.001
Arthritis	28,422 (9.37)	9031 (9.16)	0.051
Cirrhosis	6853 (2.26)	3173 (3.22)	<0.001
HIV infection	1617 (0.53)	360 (0.37)	<0.001

Abbreviations: ACP, advanced chronic patients; GMA, adjusted morbidity groups (in Spanish “Grupos de morbilidad ajustados”); CCP, complex chronic patients; HIV, human immunodeficiency virus. ^a^ Pearson’s Chi-squared test with Yates’ continuity correction.

**Table 2 ijerph-18-09473-t002:** Evolution of the sociodemographic and clinical characteristics of complex chronic patients throughout the study period, *n* (%).

	2013	2014	2015	2016	2017	2018	2019	*p*-Value ^a^
	N = 68,382	N = 70,996	N = 39,770	N = 29,098	N = 37,907	N = 33,916	N = 23,288	
**Sociodemographic characteristics**								
Sex								
Male	30,738 (45.0)	31,081 (43.8)	17,252 (43.4)	12,902 (44.3)	16,664 (44.0)	14,710 (43.4)	10,107 (43.4)	<0.001
Female	37,644 (55.0)	39,915 (56.2)	22,518 (56.6)	16,196 (55.7)	21,243 (56.0)	19,206 (56.6)	13,181 (56.6)	
Age, years								
<15	250 (0.37)	263 (0.37)	126 (0.32)	80 (0.27)	60 (0.16)	104 (0.31)	137 (0.59)	<0.001
15–44	1222 (1.79)	1596 (2.25)	593 (1.49)	453 (1.56)	534 (1.41)	364 (1.07)	298 (1.28)	
45–64	6319 (9.24)	6243 (8.79)	3109 (7.82)	2342 (8.05)	2992 (7.89)	2536 (7.48)	1925 (8.27)	
65–74	10,738 (15.7)	10,766 (15.2)	5787 (14.6)	4231 (14.5)	5714 (15.1)	4950 (14.6)	3368 (14.5)	
75–84	27,234 (39.8)	27,078 (38.1)	14,867 (37.4)	10,634 (36.5)	13,906 (36.7)	11,835 (34.9)	7941 (34.1)	
>84	22,619 (33.1)	25,050 (35.3)	15,288 (38.4)	11,358 (39.0)	14,701 (38.8)	14,127 (41.7)	9619 (41.3)	
Income level								
High	118 (0.17)	129 (0.18)	83 (0.21)	71 (0.24)	110 (0.29)	97 (0.29)	93 (0.40)	0.0000
Medium	9409 (13.8)	9663 (13.6)	5812 (14.6)	4826 (16.6)	6406 (16.9)	5886 (17.4)	4584 (19.7)	
Low	54,858 (80.2)	60,332 (85.0)	31,908 (80.2)	22,858 (78.6)	29,919 (78.9)	26,604 (78.4)	17,648 (75.8)	
Very Low	3991 (5.84)	869 (1.22)	1963 (4.94)	1343 (4.62)	1472 (3.88)	1329 (3.92)	963 (4.14)	
**Clinical characteristics**								
GMA stratification								
Baseline risk	558 (0.82)	461 (0.65)	248 (0.62)	177 (0.61)	120 (0.32)	71 (0.21)	69 (0.30)	0.0000
Low risk	6296 (9.21)	5285 (7.44)	2584 (6.50)	1628 (5.59)	1841 (4.86)	1285 (3.79)	907 (3.89)	
Moderate risk	29,499 (43.1)	29,241 (41.2)	16,310 (41.0)	11,004 (37.8)	13,834 (36.5)	11,634 (34.3)	7712 (33.1)	
High risk	32,029 (46.8)	36,009 (50.7)	20,628 (51.9)	16,289 (56.0)	22,112 (58.3)	20,926 (61.7)	14,600 (62.7)	
Comorbidities								
Diabetes mellitus	30,682 (44.9)	30,241 (42.6)	15,705 (39.5)	11,326 (38.9)	15,044 (39.7)	12,976 (38.3)	8788 (37.7)	<0.001
Heart failure	24,134 (35.3)	23,634 (33.3)	13,075 (32.9)	9554 (32.8)	11,937 (31.5)	10,672 (31.5)	7324 (31.4)	<0.001
COPD	21,552 (31.5)	21,044 (29.6)	11,076 (27.9)	8083 (27.8)	10,557 (27.8)	9164 (27.0)	6272 (26.9)	<0.001
Arterial hypertension	55,972 (81.9)	57,452 (80.9)	32,304 (81.2)	23,620 (81.2)	31,069 (82.0)	27,682 (81.6)	18,902 (81.2)	<0.001
Depression	18,709 (27.4)	20,893 (29.4)	11,895 (29.9)	9188 (31.6)	12,181 (32.1)	11,217 (33.1)	7878 (33.8)	<0.001
HIV infection	428 (0.63)	526 (0.74)	197 (0.50)	111 (0.38)	140 (0.37)	130 (0.38)	85 (0.36)	<0.001
Ischemic heart disease	17,327 (25.3)	17,116 (24.1)	8937 (22.5)	6499 (22.3)	8322 (22.0)	7189 (21.2)	4857 (20.9)	<0.001
Ictus	15,671 (22.9)	16,442 (23.2)	9478 (23.8)	7004 (24.1)	9107 (24.0)	8383 (24.7)	5762 (24.7)	<0.001
Chronic kidney disease	20,601 (30.1)	21,667 (30.5)	12,603 (31.7)	9745 (33.5)	13,558 (35.8)	12,469 (36.8)	8342 (35.8)	<0.001
Cirrhosis	1559 (2.28)	1640 (2.31)	956 (2.40)	653 (2.24)	845 (2.23)	677 (2.00)	523 (2.25)	0.016
Osteoporosis	10,591 (15.5)	12,082 (17.0)	6900 (17.3)	5260 (18.1)	7491 (19.8)	6752 (19.9)	4631 (19.9)	<0.001
Arthrosis	32,201 (47.1)	35,116 (49.5)	20,336 (51.1)	15,501 (53.3)	21,187 (55.9)	19,258 (56.8)	13,407 (57.6)	0.000
Arthritis	4927 (7.21)	6089 (8.58)	3374 (8.48)	2859 (9.83)	4221 (11.1)	3955 (11.7)	2997 (12.9)	<0.001
Dementia	7846 (11.5)	9288 (13.1)	5841 (14.7)	4424 (15.2)	6051 (16.0)	6174 (18.2)	4333 (18.6)	<0.001

Abbreviations: GMA, adjusted morbidity groups (in Spanish “Grupos de Morbilidad Ajustados”); COPD, chronic obstructive pulmonary disease; HIV, human immunodeficiency virus. ^a^ Pearson’s Chi-squared test with Yates’ continuity correction.

**Table 3 ijerph-18-09473-t003:** Evolution of the sociodemographic and clinical characteristics of advanced chronic patients throughout the study period, *n* (%).

	2013	2014	2015	2016	2017	2018	2019	*p*-Value ^a^
	N = 13,206	N = 18,137	N = 14,755	N = 12,918	N = 14,587	N = 14,040	N = 10,944	
**Sociodemographic characteristics**								
Sex								
Male	6122 (46.4)	8440 (46.5)	6950 (47.1)	6134 (47.5)	6828 (46.8)	6495 (46.3)	5038 (46.0)	0.236
Female	7084 (53.6)	9697 (53.5)	7805 (52.9)	6784 (52.5)	7759 (53.2)	7545 (53.7)	5906 (54.0)	
Age, years								
<15	31 (0.23)	18 (0.10)	15 (0.10)	13 (0.10)	11 (0.08)	27 (0.19)	137 (0.59)	<0.001
15–44	208 (1.58)	213 (1.17)	158 (1.07)	126 (0.98)	152 (1.04)	123 (0.88)	298 (1.28)	
45–64	1310 (9.92)	1623 (8.95)	1292 (8.76)	1167 (9.03)	1333 (9.14)	1286 (9.16)	1925 (8.27)	
65–74	1694 (12.8)	2380 (13.1)	1944 (13.2)	1720 (13.3)	1894 (13.0)	1840 (13.1)	3368 (14.5)	
75–84	4306 (32.6)	5557 (30.6)	4439 (30.1)	3606 (27.9)	4107 (28.2)	3796 (27.0)	7941 (34.1)	
>84	5657 (42.8)	8346 (46.0)	6907 (46.8)	6286 (48.7)	7090 (48.6)	6968 (49.6)	9619 (41.3)	
Income level								
High	40 (0.30)	45 (0.25)	42 (0.28)	45 (0.35)	52 (0.36)	66 (0.47)	19 (0.17)	<0.001
Medium	1829 (13.9)	2563 (14.1)	2310 (15.7)	2294 (17.8)	2717 (18.6)	2600 (18.5)	94 (0.86)	
Low	10,622 (80.5)	15,284 (84.3)	11,791 (79.9)	10,110 (78.3)	1130 (77.5)	10,893 (77.6)	910 (8.32)	
Very Low	708 (5.36)	244 (1.35)	609 (4.13)	469 (3.63)	517 (3.54)	481 (3.43)	1399 (12.8)	
**Clinical characteristics**								
GMA stratification								
Baseline risk	88 (0.67)	65 (0.36)	60 (0.41)	33 (0.26)	26 (0.18)	19 (0.14)	13 (0.12)	0.0000
Low risk	840 (6.36)	888 (4.90)	615 (4.17)	378 (2.93)	331 (2.27)	276 (1.97)	180 (1.64)	
Moderate risk	4562 (34.5)	5627 (31.0)	4258 (28.9)	3393 (26.3)	3485 (23.9)	3015 (21.5)	2172 (19.8)	
High risk	7716 (58.4)	11,557 (63.7)	9822 (66.6)	9114 (70.6)	10,745 (73.7)	10,730 (76.4)	8579 (78.4)	
Comorbidities								
Diabetes mellitus	4963 (37.6)	6627 (36.5)	5262 (35.7)	4646 (36.0)	5350 (36.7)	4983 (35.5)	4017 (36.7)	0.005
Heart failure	4768 (36.1)	6390 (35.2)	5204 (35.3)	4539 (35.1)	5221 (35.8)	4891 (34.8)	4071 (37.2)	0.002
COPD	3992 (30.2)	5590 (30.8)	4522 (30.6)	3861 (29.9)	4428 (30.4)	4144 (29.5)	3417 (31.2)	0.053
Arterial hypertension	10,060 (76.2)	13,823 (76.2)	11,290 (76.5)	10,048 (77.8)	11,471 (78.6)	10,923 (77.8)	8669 (79.2)	<0.001
Depression	3304 (25.0)	4835 (26.7)	4098 (27.8)	3684 (28.5)	4460 (30.6)	340 (30.9)	3584 (32.7)	<0.001
HIV infection	63 (0.48)	68 (0.37)	55 (0.37)	41 (0.32)	48 (0.33)	45 (0.32)	40 (0.37)	0.348
Ischemic heart disease	3090 (23.4)	4031 (22.2)	3086 (20.9)	2678 (20.7)	3089 (21.2)	2917 (20.8)	2296 (21.0)	<0.001
Ictus	3122 (23.6)	4574 (25.2)	3771 (25.6)	3301 (25.6)	3963 (27.2)	3887 (27.7)	3117 (28.5)	<0.001
Chronic kidney disease	4121 (31.2)	5778 (31.9)	4855 (32.9)	4535 (35.1)	5485 (37.6)	5381 (38.3)	4384 (40.1)	<0.001
Cirrhosis	425 (3.22)	616 (3.40)	495 (3.35)	420 (3.25)	506 (3.47)	398 (2.83)	313 (2.86)	0.010
Osteoporosis	1839 (13.9)	2823 (15.6)	2317 (15.7)	2163 (16.7)	2658 (18.2)	2604 (18.5)	2168 (19.8)	<0.001
Arthrosis	5556 (42.1)	7966 (43.9)	6863 (46.5)	6275 (48.6)	7452 (51.1)	7286 (51.9)	5922 (54.1)	<0.001
Arthritis	914 (6.92)	1352 (7.45)	1212 (8.21)	1219 (9.44)	1506 (10.3)	1474 (10.5)	1354 (12.4)	<0.001
Dementia	2050 (15.5)	3257 (18.0)	2761 (18.7)	2581 (20.0)	3172 (21.7)	3766 (26.8)	2905 (26.5)	<0.001

Abbreviations: GMA, adjusted morbidity groups (in Spanish “Grupos de Morbilidad Ajustados”); COPD, chronic obstructive pulmonary disease; HIV, human immunodeficiency virus. ^a^ Pearson’s Chi-squared test with Yates’ continuity correction.

**Table 4 ijerph-18-09473-t004:** Prevalence rates of complex chronic and advanced chronic patients according to income level (year 2019), rate per 1000 people.

	CCP		ACP	
	Women	Men	Women	Men
**Income level, €/year**				
High (>100,000)	8.0	6.0	1.6	1.4
Intermediate (18,000–100,000)	12.0	13.1	2.1	2.3
Low (<18,000)	36.4	27.0	6.2	4.6
Very low (unemployed/receiving welfare support)	36.8	25.7	5.1	3.9

Abbreviations: ACP, advanced chronic patients; CCP, complex chronic patients.

**Table 5 ijerph-18-09473-t005:** Main comorbidities in complex chronic and advanced chronic patients and their corresponding non-CCP and non-ACP populations adjusted by age, sex, and annual income (year 2019), %.

	CCP	Adjusted Non-CCP Population	*p*-Value ^a^	ACP	Adjusted Non-ACP Population	*p*-Value ^a^
Diabetes	43.8	24.4	<0.001	38.3	26.6	<0.001
Chronic kidney disease	41.5	24.2	<0.001	42.0	29.2	<0.001
Heart failure	39.0	13.8	<0.001	40.7	20.0	<0.001
Cancer	34.2	25.0	<0.001	48.5	26.4	<0.001
COPD	32.3	15.3	<0.001	32.5	18.0	<0.001
Dementia	30.4	13.9	<0.001	38.8	18.2	<0.001
Stroke	29.3	13.9	<0.001	31.2	17.0	<0.001
Ischemic heart disease	26.1	12.7	<0.001	24.1	15.3	<0.001
Arthritis	14.7	9.4	<0.001	13.0	10.1	<0.001
Asthma	12.8	7.2	<0.001	11.3	7.9	<0.001
Alcoholism	5.9	2.0	<0.001	5.9	2.2	<0.001
Atypical psychosis	4.6	1.6	<0.001	5.1	2.2	<0.001
Major depressive disorder	4.2	2.0	<0.001	3.4	2.1	<0.001
Cirrhosis	2.8	0.9	<0.001	3.4	1.0	<0.001
Schizophrenia	1.8	0.6	<0.001	1.3	0.7	<0.001
Bipolar disorder	1.4	0.6	<0.001	-	-	<0.001

Abbreviations: ACP, advanced chronic patients; CCP, complex chronic patients; COPD, chronic obstructive pulmonary disease. ^a^ Rate ratio by median-unbiased estimation (mid-p).

**Table 6 ijerph-18-09473-t006:** Utilization of healthcare services by complex chronic (CCPs) and advanced chronic (ACPs) patients and their corresponding non-CCP and non-ACP populations adjusted by age, sex, and annual income, and their associated expenditures (year 2019).

	CCP	Adjusted Non-CCP Population	*p*-Value ^a^	ACP	Adjusted Non-ACP Population	*p*-Value ^a^
**Healthcare services utilization**						
Ambulatory healthcare services (visits or admissions per patient and year), mean						
Primary care	21.1	11.3	<0.001	22.2	12.8	<0.001
Outpatient care	4.3	2.6	<0.001	4.7	2.5	<0.001
Emergency department	1.3	0.6	<0.001	1.6	0.7	<0.001
Day hospital	0.7	0.2	<0.001	1.5	0.3	<0.001
Mental health	0.2	0.1	<0.001	0.1	0.1	<0.001
Prescribed drugs (number per patient and year)	12.6	8.0	<0.001	12.7	8.7	<0.001
Rate of admissions (institutionalizations), admissions per 100 patients and year						
Acute care hospital	64.4	27.1	<0.001	88.4	31.9	<0.001
Intermediate care hospital	17.0	5.7	<0.001	35.5	8.1	<0.001
Psychiatric center	0.5	0.1	<0.001	0.2	0.1	<0.001
**Healthcare services expenditure (€ per person and year) (%) ^b^**			***p*-value ^c^**			***p*-value ^c^**
Primary care	653.5 (10.75)	367.8 (14.98)	<0.001	667.3 (8.35)	413.5 (14.59)	<0.001
Outpatient care	441.2 (7.26)	225.1 (9.17)	<0.001	618.1 (7.73)	221.9 (7.83)	<0.001
Hospital admissions	1713.6 (28.19)	698.8 (28.46)	<0.001	2385.9 (29.84)	821.1 (28.98)	<0.001
Emergency department	551.4 (9.07)	223.9 (9.12)	<0.001	696.3 (8.71)	286.0 (10.09)	<0.001
Mental health	30.8 (0.51)	10.4 (0.42)	<0.001	11.5 (0.14)	9.5 (0.34)	0.167
Intermediate care center	475.9 (7.83)	163.5 (6.66)	<0.001	774.4 (9.69)	224.4 (7.92)	<0.001
Prescribed drugs	1709.2 (28.12)	684.9 (27.90)	<0.001	2211.6 (27.66)	742.4 (26.20)	<0.001
Other healthcare services	502.8 (8.27)	80.9 (3.30)	<0.001	630.1 (7.88)	114.4 (4.04)	<0.001
**Total healthcare costs**	6078.3	2455.2	<0.001	7995.2	2833.3	<0.001

Abbreviations: ACPs, advanced chronic patients; CCPs, complex chronic patients. ^a^ Rate ratio test by median-unbiased estimation (mid-p); ^b^ Calculated over the total healthcare costs for each group; ^c^ Student’s *t*-test.

## Data Availability

The datasets used and/or analyzed during the current study are available from the corresponding author upon reasonable request.

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
