# Peer review of "Characteristics and Service Utilization by Complex Chronic and Advanced Chronic Patients in Catalonia: A Retrospective Seven-Year Cohort-Based Study of an Implemented Chronic Care Program"

_ijerph, 2021, doi:10.3390/ijerph18189473_

Round 1
Reviewer 1 Report
Comments for authors:
Table 1 – authors should compare baseline characteristics between ACP and CCP group.
Line 112 – NECPAL tool for ACP identification should be explained in more details
Table 2 and Table 3 – authors should specify which test does the P-value refers to in the footnote of table.
Figure 6 – please add an arrow so that is clear on which color does the percentages refer to
Line 289, Table 5 and line 303 and Table 6 – authors should perform or add standard statistical analysis in order to compare variables between CCP and nonCCP, and ACP and nonACP population, respectively. In fact, it is not clear how did you even compare these groups as no statistical analysis was presented.
Line 330 – “CCPs’ and ACPs’ cumulative incidence for the 2017-2019 period in the different regions of Catalonia was adjusted to population’s age, sex, morbidity and income level, revealing regions with increased and decreased case incidence compared to the expected rates (Figure 7).” Why this analysis wasn’t performed for the whole study duration but only from 2017-2019?
Discussion:
- Authors should interpret the reduction in incidence and increase in prevalence observed for both CCP and ACP.
- Authors should find similar chronic care programs tested in different or similar populations and compare them to their study results
Reviewer 2 Report
I read this paper with great interests. Authors reviewed all CCPs and ACPs identified Catalonia and extracted characteristics in these pain possessors. They clarified CCPs’ and ACPs’ prevalence increased and 26
was higher in lower-income populations; most cases were women. CCPs and ACPs had all comorbidities at higher frequencies, higher utilization of healthcare services, and were more frequently at high risk (63% and 71%, respectively) than age-, sex-, and income level-adjusted non-CCP (23%) and non-ACP populations (30%). I totally agree with these tendencies in the population. I would like to recommend one point. In Discussion section, they should describe the difference between advanced countries and developing countries.
Author Response
Manuscript ID ijerph-1296675
Responses to reviewer and actions taken
Response to reviewer #2
The authors would like to thank the receipt of reviewers’ comments, which will certainly improve the quality of the manuscript. Please, find a point-by-point response to the reviewer’s comments below.
Comment 1. In Discussion section, they should describe the difference between advanced countries and developing countries.
Response: As requested by the reviewer, we have discussed the results of this study and their implications in developing and low-income countries by including additional text in the discussion section of the revised version of the manuscript (page 19, lines 1419-1420 and page 20, lines 1446-1448). In addition, regarding patients’ classification based on income level, we have clarified the criteria used for classification in the discussion section of the revised version of the manuscript (page 19, lines 1387-1389).
Round 2
Reviewer 1 Report
No further comments.